# Single-cell genomics of multiple uncultured stramenopiles reveals underestimated functional diversity across oceans

Yoann Seeleuthner et al.[#]

Single-celled eukaryotes (protists) are critical players in global biogeochemical cycling of nutrients and energy in the oceans. While their roles as primary producers and grazers are well appreciated, other aspects of their life histories remain obscure due to challenges in culturing and sequencing their natural diversity. Here, we exploit single-cell genomics and metagenomics data from the circumglobal *Tara* Oceans expedition to analyze the genome content and apparent oceanic distribution of seven prevalent lineages of uncultured heterotrophic stramenopiles. Based on the available data, each sequenced genome or genotype appears to have a specific oceanic distribution, principally correlated with water temperature and depth. The genome content provides hypotheses for specialization in terms of cell motility, food spectra, and trophic stages, including the potential impact on their lifestyles of horizontal gene transfer from prokaryotes. Our results support the idea that prominent heterotrophic marine protists perform diverse functions in ocean ecology.

#A full list of authors and their affliations appears at the end of the paper

The microbial loop in planktonic ecosystems is the process by which suspended organic matter produced within food webs is channeled through heterotrophic prokaryotes and their tiny grazers and eventually transferred to higher trophic levels or remineralized[1]. Very small but numerous marine heterotrophic protists play key roles in these processes. Since most of them remain uncultured, their functions remain largely unknown[2]. A recent DNA metabarcoding survey based on *Tara* Oceans global plankton samples has revealed the existence of thousands of heterotrophic protist taxa in eukaryotic communities[3] that potentially participate in numerous species interaction networks in yet-to-be defined ways[4]. An extensive genome-level description of abundant marine heterotrophic protists could therefore be a key step toward understanding their ecological roles. Currently, the only way to obtain such information is through single-cell sequencing, although the technology is still in its infancy for eukaryotic cells[5–10], since generated assemblies are highly fragmented and rarely complete.

Here, we integrate single-cell genomics with metagenomic and metatranscriptomic sequence data for exploring the ecological and functional complexity of uncultured micro-eukaryotes, key players in the world's largest ecosystem. We selected for our study 40 single cells representative of three uncultured stramenopile clades that are known to be abundant in marine pico-nano plankton. Marine stramenopile group 4 (MAST-4) representatives are small, flagellated, bacterivorous cells that are abundant in temperate and tropical oceans[11,12]. A partial genome of a MAST-4 clade D was previously characterized using single-cell sequencing[8]. In this study, we present three distinct genomes from clades A, C, and E, clearly divergent from clade D. MAST-3[11] is a very diverse group of small flagellated organisms that includes a potential diatom epibiont and one cultured strain[13,14]. Heterotrophic chrysophytes from the Clade H additionally appear to be abundant in the ocean, according to environmental DNA surveys[15]. It has been postulated that all of these lineages originated from a presumably autotrophic stramenopile ancestor[16], although lack of genome information has hindered understanding of the evolution of heterotrophy vs. autotrophy within the stramenopiles. Assessment of the genes involved in the degradation of organic matter may thus be relevant for elucidating their roles in marine ecosystems and biogeochemical cycles[17].

## Results

**Assembly strategy.** More than 900 single-cell amplified genomes (SAGs) were generated from small heterotrophic protists selected from eight *Tara* Oceans sampling stations representing contrasting environments in the Mediterranean Sea and Indian Ocean. SAGs belonging to the target lineages were identified by PCR and subsequent sequencing of their 18S rRNA gene. A total

of 40 SAGs were sequenced[18]: 23 from three MAST-4 lineages (MAST-4A, MAST-C, and MAST-E), six from two lineages of MAST-3 (MAST-3A and MAST-F), and 11 from two lineages of chrysophytes (Chrysophytes H1 and H2). We also generated metagenomic and metatranscriptomic datasets from the 0.8 to 5 μm size fraction collected from 76 and 68 *Tara* Oceans sampling sites, respectively, to assist the removal of potential contaminants from nuclear sequences and to improve gene structures (see section "Methods"; Supplementary Fig. 1, and companion papers[18,19]). The characteristics of each composite genome are summarized in Table 1. The MAST-4A cells were co-assembled as two independent sets of sequences, for use as an internal control for subsequent analyses and because they originated from two different water masses; however, they were very similar in genome composition (Supplementary Fig. 2) and a single assembly would have been possible[20].

**Functional repertoires.** To assess variation in the functional repertoires of the sequenced uncultured stramenopiles and to provide further context, we predicted functional domains (Pfams) in each annotated protein from each of the lineages, and compared their diversity and abundance against each other and against other sequenced stramenopile genomes. We then calculated pairwise distances between genomes based on relative Pfam abundances. The resulting pattern (Fig. 1a) indicated that the uncultured heterotrophic stramenopiles contained a diversity of gene repertoires, comparable to those of the sequenced genomes of autotrophic stramenopiles. However, the composition of each genome clustered primarily according to the trophic mode of each organism, with groups corresponding to heterotrophs, single-celled autotrophs, multicellular autotrophs, and mixotrophs. Moreover, within the heterotrophs, the MAST lineages and the chrysophytes-clade H clustered into a single functional group despite their distant phylogenetic positions (Fig. 1b, Supplementary Fig. 3). They could also be clearly distinguished from the plant-parasitic and gut-commensal heterotrophic stramenopiles (Fig. 1a, groups 3, 5, and 6), suggesting ecosystem-specific functional diversification, which needs further investigation.

Within the marine SAG genomes, many gene families showed differential abundances, indicating that functional capacities are distinct (Supplementary Table 1). One extreme pattern was observed for genes encoding the axonemal dynein heavy chain (DHC), which is an essential flagellar component. Almost all SAG genomes contained a family of genes encoding DHCs, with the exception of MAST-3A, for which we could not detect a single full-length gene and observed a significant decrease in the number of DHC Pfam domains (Supplementary Fig. 4). A closer examination of the MAST-3A genome regions containing the DHC-associated Pfam domains showed evidence of advanced

**Table 1 SAGs assembly and annotation summary**

| Name | Number of cells | Raw assembly size (Mbp) | Cross SAG sequences (Mbp) | Outlier sequences (Mbp) | Final assembly size (Mbp) | N50 | BUSCO v2 complete genes (%) | Number of predicted genes |
|---|---|---|---|---|---|---|---|---|
| Chrysophyte H1 | 8 | 16.7 | 0.1 | 0.6 | 15.9 | 25,581 | 57 | 3050 |
| Chrysophyte H2 | 3 | 14.3 | 1.1 | 0.3 | 10.6 | 10,194 | 27 | 1637 |
| MAST-3A | 4 | 20.0 | 0 | 1.0 | 18.9 | 6223 | 53 | 3289 |
| MAST-3F | 2 | 21.5 | 0 | 0.3 | 21.1 | 7132 | 37 | 2694 |
| MAST-4A1 | 6 | 33.4 | 0 | 1.0 | 31.8 | 10,950 | 59 | 8018 |
| MAST-4A2 | 4 | 37.1 | 3.0 | 1.1 | 32.8 | 11,577 | 64 | 8537 |
| MAST-4C | 4 | 31.2 | 0 | 0.9 | 30.0 | 8097 | 54 | 5478 |
| MAST-4E | 9 | 30.3 | 0.2 | 1.4 | 28.4 | 9788 | 61 | 4652 |

*SAG* single amplified genome, *N50* length of the shortest scaffold from the minimal set of scaffolds representing 50% of the assembly size, *BUSCO v2* number of complete genes found using the BUSCO program (Benchmarking Universal Single-Copy Orthologs)

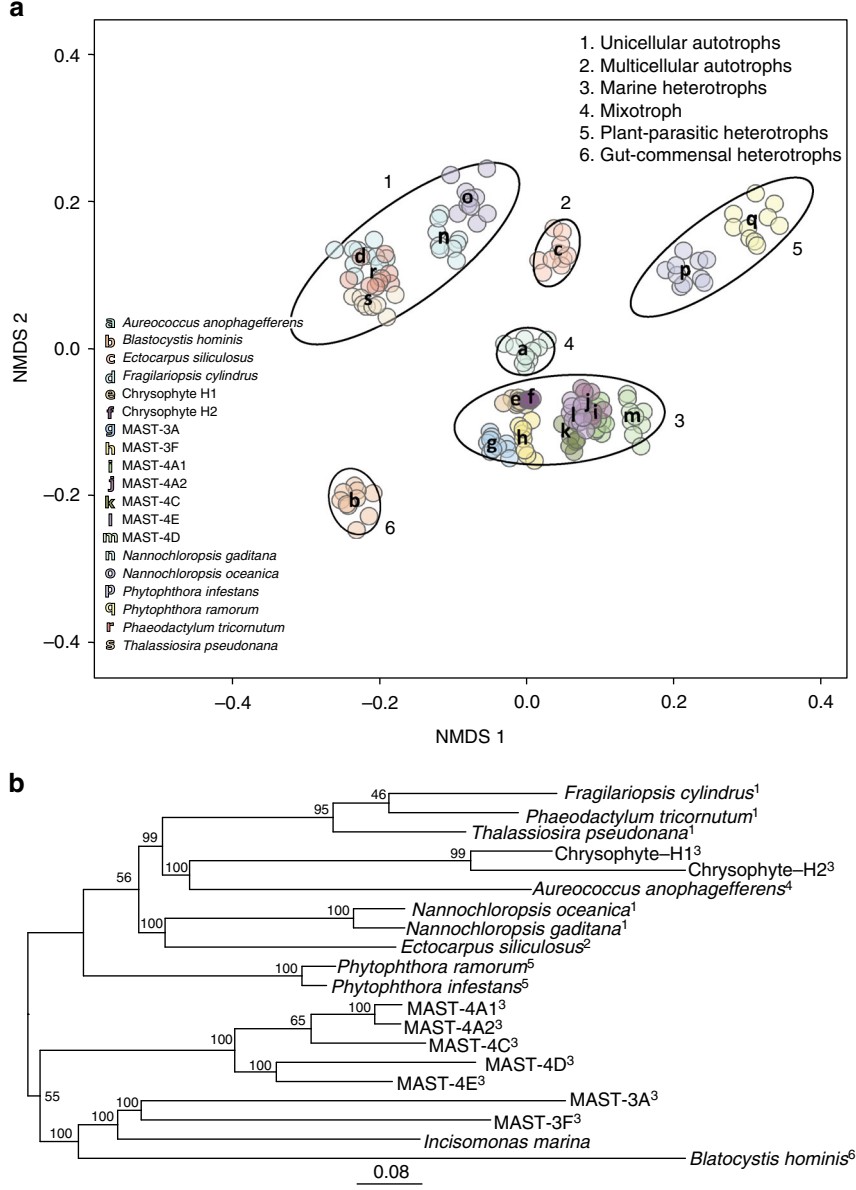

**Fig. 1** Marine heterotrophic SAG lineages form a functional group distinct from autotrophs and other heterotrophs. **a** Non-metric multidimensional scaling (NMDS) projection of a Bray–Curtis distance matrix that shows Pfam motif occurrences in various stramenopile genomes. Because the genome sequences are incomplete, a rarefying procedure was applied to obtain 1400 Pfam motifs per genome. Ten independent rarefied samples were obtained and used for NMDS. Ellipses (at 95% confidence limit) were drawn by using the 'ordiellipse' function of the vegan package in R, with the group defined by life history mode (indicated by number in top right). Letters indicate the positions of the mean coordinates of the 10 rarefied Pfam counts per organism. The analysis was conducted on 19 stramenopile genomes, which included MAST-4D[8]. Marine heterotrophic stramenopiles from this study form a large but coherent group (Group 3), which is distinct from autotrophic species and heterotrophic species from other environments. **b** Phylogenetic tree from the analysis of a total of 160 conserved eukaryotic proteins using maximum likelihood. Protein sequences of *Incisomonas marina* from ref.[44] are included. Indices indicate life history mode as in panel **a**. Bootstrap values are represented on internal nodes. The branch length represents the mean number of substitutions per site

pseudogenization (Supplementary Fig. 4c and e–i), indicating that relatively recent gene loss events are responsible for the absence of DHC-encoding genes. Although we did not observe DHC reduction in the MAST-3F genome (Supplementary Table 1; Supplementary Fig. 4a), previous morphological analyses of other MAST-3 members had indicated reduced motility and the presence of only a single flagellum[12,13]. *Solenicola setigera* (MAST-3I clade) is found living epiphytically on diatoms, while the cultured *Incisomonas marina* (MAST-3J clade) seems to be a bad swimmer, with cells generally attaching to surfaces. Motility may therefore have been dispensed with on multiple occasions in

these organisms, and may be congruent with the switch to epiphytic or parasitic lifestyles in several MAST-3 lineages.

We further observed the presence of rhodopsin coding genes exclusively in the MAST-4C lineage, suggesting again functional adaptation. Two rhodopsin classes with distinct functions are known: sensory rhodopsins act as light sensors for diverse signal transduction pathways, whereas proteorhodopsins are light-driven proton pumps that synthesize ATP independently of photosynthesis[21]. Phylogenetic analysis of these two rhodopsin genes revealed that they are related to previously described proteorhodopsins of diatoms, dinoflagellates and haptophytes,

and are evolutionarily distant from prokaryotic proteorhodopsins[22,23] (Supplementary Fig. 5). MAST-4C rhodopsins are thus eukaryotic proteorhodopsins, not derived from recent bacterial gene transfers. No proteorhodopsins were found in the other lineages, suggesting a specific genetic adaptation of MAST-4C to phototrophy. The MAST-4C proteorhodopsin genes appear to be highly expressed in surface samples, representing more than 3% of the total MAST-4C transcripts (Supplementary Fig. 5b). We further observed that MAST-4C cells were preferentially detected in samples from tropical surface waters (see below).

We then explored the gene families related to organic carbon acquisition in the various MAST lineages, and used Carbohydrate-active enzymes (CAZymes) as indicators of nutrient acquisition and more generally of organismal glycobiological potential[24]. The CAZyme-encoding gene profiles indicated a large repertoire of glycoside hydrolases (GHs) in almost all genomes, with many bearing secretion peptide signals (Supplementary Table 2). This is consistent with the bacterivorous lifestyle proposed for most of these organisms, which have the capacity to degrade bacterial carbohydrates and to target them for degradation in phagosomes. MAST-4 was found to be the most CAZyme-rich group, consistent with it including only bacterivorous lineages. On the other hand, MAST-3F appears to have a very limited CAZyme repertoire, almost none of which appear to be secreted. The MAST-3F genome also encodes fewer hydrolytic enzymes of other types, such as proteases (Supplementary Table 1), indicating that MAST-3F may not be bacterivorous. The other most CAZyme-poor genomes are those of chrysophytes, a group containing many photosynthetic organisms with mixotrophic behavior. This suggests complex evolutionary patterns in chrysophyte genomes, with intricate losses and/or gains of genes involved in photosynthesis and heterotrophy.

Putative substrates were predicted on all encoded CAZymes theoretically capable of cleaving complex carbohydrates (GHs and polysaccharide lyases) to reveal which enzymes are involved in bacterivory and possible carbohydrate acquisition from other sources (Fig. 2). Identification of lysozymes from the GH25 family in most co-assembled genomes could be indicative of peptidoglycan breakdown. Moreover, in all MAST-4 and MAST-3A genomes, suites of genes encoding enzymes able to hydrolyze all the components of green and brown algal cell walls were detected, including cellulose, xylan, pectin, and agarose (Fig. 2). Interestingly, examination of sequences that were considered as contaminants during genome reconstruction revealed large fragments of chloroplast, and sometimes even nuclear, DNA from photosynthetic eukaryotes in two of the MAST-4A and one of the MAST-4E cells, but not in any of the other lineages (Supplementary Table 2). MAST-4 was previously shown to have the capacity to ingest eukaryotic microalgae in an experimental setting in the presence of high algal concentrations[25]. Our observations provide further evidence for the role of MAST-4 and MAST-3A in algal consumption, which could have a significant impact on the transfer of organic material from primary producers to higher trophic levels. Further function predictions identified candidate secreted enzymes for the breakdown of starch, chitin, and beta-1,3-glucans (Fig. 2). The above observations imply that the examined organisms may have the capacity to degrade organic materials from bacteria and algae, as well as from chitin-containing organisms, such as fungi, diatoms, and crustaceans, emphasizing their global involvement and differentiated roles in the microbial loop.

For the MAST-4A, MAST-4C, MAST-4E, and MAST-3A genomes, the number of GH genes exceeded that of glycosyltransferases (GTs), with the GH/GT ratio ranging from 1.6 to 2, reflecting the heterotrophic nature of these organisms. However, the MAST-3F and chrysophytes H1 and H2 genomes displayed higher numbers of GTs than GHs, indicating that these organisms may be less dependent on carbohydrate degradation.

**Horizontally transferred genes**. Another fundamental question is whether heterotrophic protists are impacted by horizontal gene transfer (HGT) from the prey they ingest. We assessed the extent to which genes had probably been acquired by horizontal transfer from prokaryotes in each SAG lineage (see section "Methods"). The proportion of potential HGT events was different among the studied genomes (Supplementary Table 3). The lowest observed value was for MAST-3F, which was also the genome lacking elements suggestive of a bacterivorous lifestyle (see above). A link could therefore exist between bacterivory and prokaryotic gene acquisition in the other lineages. Furthermore, the functional classification of candidate HGTs based on Clusters of Orthologous Groups (COGs)[26] showed a bias towards metabolic activities (Supplementary Fig. 6a and 6b). Refining the metabolic COG categories revealed an even more pronounced bias towards activities linked to carbohydrate and protein degradation, defense/resistance against bacteria and nitrogen utilization (Supplementary Table 4). Overall, our data indicate that each MAST lineage may have a different functional profile in terms of organic matter processing, and that HGT may have contributed to enabling this metabolic specialization.

**Geographical distributions**. Finally, we used metagenomic fragment recruitment from the 0.8 to 5 μm size-fraction of the *Tara* Oceans metagenomics dataset to explore the global distribution of the studied lineages and of MAST-4 D (Fig. 3). In addition to quantifying lineage-specific abundances, metagenomics data was used to obtain indications of genetic diversification by using the similarity of nucleotide sequences to each reference genome as a measure of divergence (Supplementary Fig. 7). Widely differing geographic distributions were observed. First, the previously sequenced MAST-4 D genome is encountered in only one coastal sample from the South Atlantic Ocean, indicating that open ocean populations of MASTs can differ from coastal ones. In the studied lineages, only one organism with a well-conserved genotype, MAST-4A, appears to be cosmopolitan, although it was not detected in the Southern Ocean. Another group, MAST-4C, displays high genetic homogeneity worldwide but with a geographic range restricted mostly to tropical and subtropical waters, except in the sub-tropical Atlantic Ocean. In other cases, we observed the existence of genotype subsets divergent from the reference genomes, with preferential geographic patterns (MAST-4E, MAST-3A, and chrysophyte H1). Finally, chrysophyte H2 and MAST-3F are low-abundance species encountered in different regions as divergent genotypes.

Each of the distributions was compared to the environmental parameters recorded at each sampling site[27,28] (the four most significant parameters are highlighted in Supplementary Fig. 8). The most significant parameter that discriminate the distributions (Kruskal–Wallis test $p$-value = $2.2 \times 10^{-16}$) was water temperature (Fig. 4a), suggesting that some of these species likely have preferential temperature ranges in which they are maximally abundant. Divergent MAST-3A and MAST-4E genomes were found in water temperatures distinct from where organisms with genomes more similar to the reference SAG genome thrive (Wilcoxon test, $p$-value $< 2 \times 10^{-2}$ and $p$-value $< 3 \times 10^{-4}$, respectively; Fig. 4b and c). Finally, depth-dependent distributions were also frequent, with MAST-4C and MAST-3A being located preferentially in the subsurface, while MAST-4E and Chrysophyte H1 were found predominantly at the deep chlorophyll maximum (DCM), except in well-mixed water columns (Fig. 3).

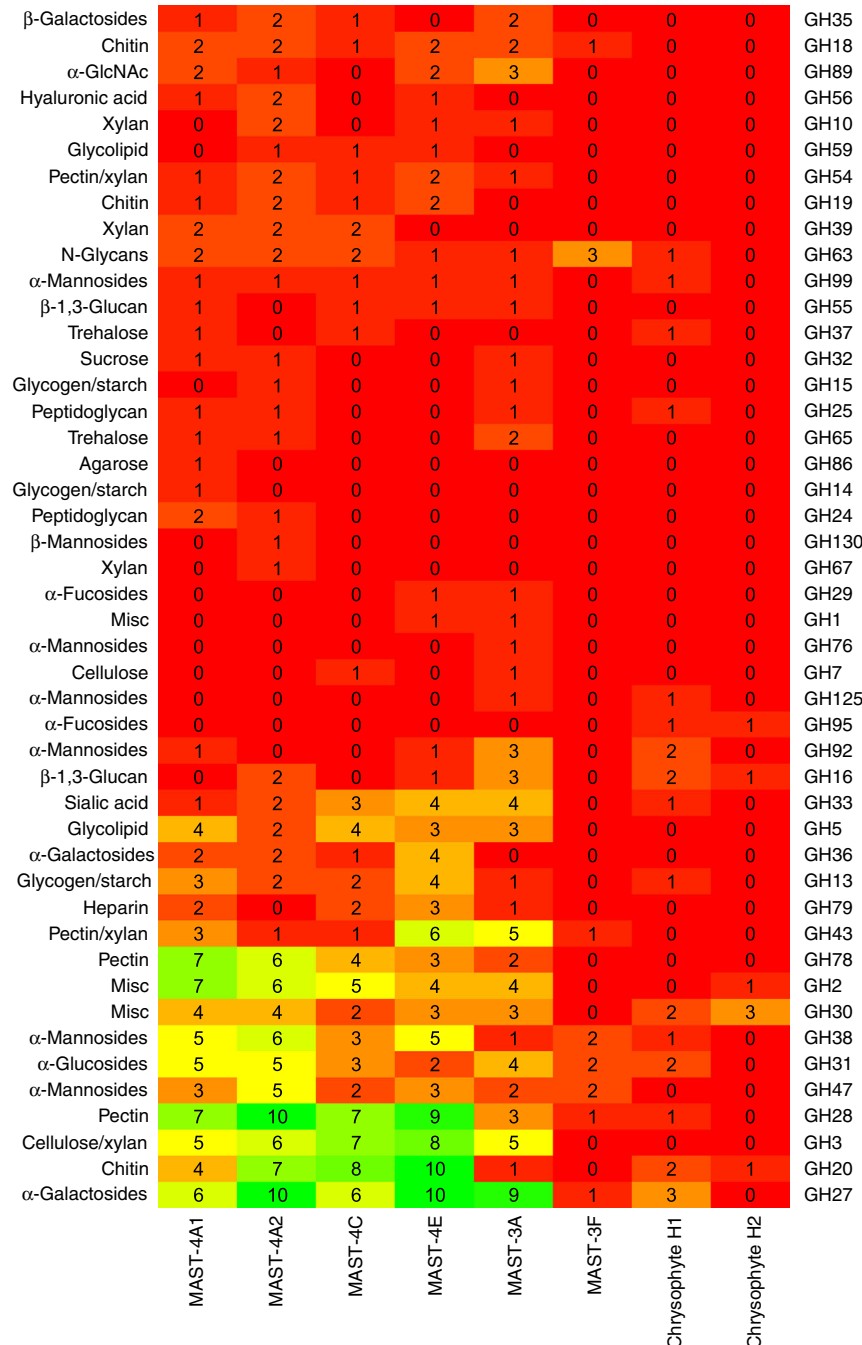

**Fig. 2** SAG lineage glycoside hydrolases (GHs). GH families are numbered (right) according to the CAZyme database. Potential substrates are indicated on the left side. Internal numbers represent the number of genes in each genome predicted to belong to the GH category. Colors indicate the number of predicted GH genes per family, from low (red) to high (green)

## Discussion

Our findings indicate that each of the examined taxa may have a specific spatial distribution that correlates with environmental parameters, principally ocean provinces, temperature, and depth. However, some limitations of the data set—mostly its single time point per location, the use of *Tara* Oceans metagenomes as the only resource, the relatively low resolution of sampling points per geographical area, and the absence of metagenomics replicates— may have under-estimated the true distribution of the organisms studied here. Notwithstanding, the *Tara* Oceans data set is by far the largest available today, and is the only extensive metage-nomics effort tackling specifically the size fraction where these heterotrophic protists can be found (no additional location was

revealed using the other available size fractions). The relatively low resolution of sampling locations is balanced by a careful choice of oceanographic situations in each sampled region. The depth of sequencing is also particularly significant compared to other studies (at about 25 Gb per sample), so the use of replicates will be of low utility for detecting the presence of the genomes under study here. The major limitation in our view is the absence of temporal information from each sampling location. Although *Tara* Oceans was a 3-year expedition that sampled plankton across all seasons, each location is currently described at a single time only and so it will be interesting to extend our results in future sampling campaigns by targeting sites of interest during different seasons.

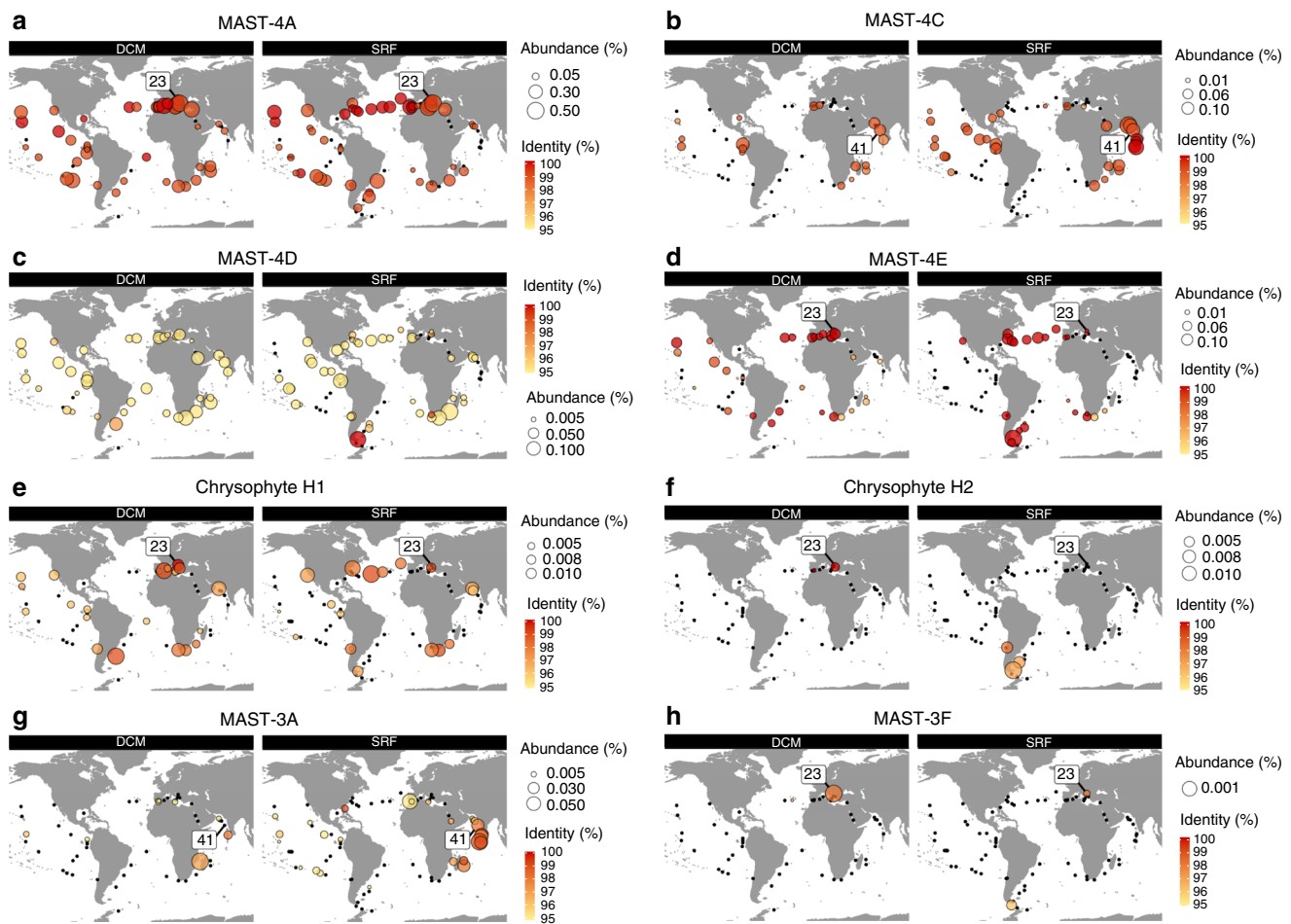

**Fig. 3** Biogeographic distribution of the SAG lineages based on metagenome read recruitment with separation between deep chlorophyll maximum and subsurface. Global maps showing the presence of the SAG lineages based on metagenomics read mapping at each *Tara* Oceans station either as a black dot (no signal detected) or as a circle whose diameter indicates the species relative abundance. Abundance in samples from deep chlorophyll maximum (DCM, left panel) often differs from surface samples (SRF, right panel): only MAST-4A shows the same pattern in DCM and SRF samples (**a**). The color inside each circle provides the median percentage similarity of the reads to the reference. The station from where the SAG originates is indicated by its number. **a** MAST-4A; **b** MAST-4C; **c** MAST-4D; **d** MAST-4E; **e** Chrysophyte H1; **f** Chrysophyte H2; **g** MAST-3A; and **h** MAST-3F

Moreover, the differentiated gene content between taxa suggests specific distinctive functional capacities even within taxa. This indicates that, like prokaryotes and phytoplankton[29–31], heterotrophic protists are not interchangeable components of marine plankton ecosystems, but effectively participate from varied perspectives in the highly complex networks of interacting taxa[4,32].

## Methods

**Single-cell isolation and amplification**. Aquatic samples were collected during the *Tara* Oceans expedition[23,33]. One-milliliter aliquots were amended with 6% (final concentration) glycine betaine and stored at −80 °C[34]. Flow-cytometric sorting, whole genome amplification, and sequencing of partial 18S rRNA genes of single cells were performed by the Bigelow Laboratory Single Cell Genomics Center (https://scgc.bigelow.org/), following previously described protocols[5,7] with a slight modification: 1x SYBR Green I (Life Technologies Corporation) was used instead of Lysotracker Green to stain the cells[18]. The 40 SAGs analyzed in this study came from the Mediterranean Sea (sampled in November 2009) and Indian Ocean (sampled in March 2010) (Table 1). Cell sorting was performed on cells lacking chlorophyll. Therefore all cells were considered heterotrophic.

**Sequencing and assembly**. The steps used for assembly, annotation, and contamination control are summarized in Supplementary Fig. 1a. Library preparation from single cells is described in Alberti et al.[18]. All cells were independently sequenced on a 1⁄8th Illumina HiSeq lane, which produced ~25 million 101-bp paired-end reads. Reads from SAGs with highly similar 18S were first co-assembled using the HyDA assembler[35]. Based on colored de Bruijn graphs, HyDA outputs

the contribution of each library to each contig, which provides a criterion to determine which libraries can be co-assembled: only libraries that cover a large fraction of the longest contigs were pooled, which ensured that the genomes were close enough to be co-assembled. Libraries that were successfully co-assembled with HyDA were then re-assembled using SPAdes 2.4[36], which provided the best results in terms of assembly size, N50 and number of core eukaryotic genes recovered. Although SPAdes provides an integrated scaffolder, we re-scaffolded contigs with SSPACE v2[37] and filled gaps with GapCloser (SOAPdeNovo2 package [v 1.12-6][38]). Scaffolds shorter than 500 bp were discarded from the assembly. Accession numbers of generated assemblies can be found in Supplementary Table 5.

**Removal of organelle sequences**. Because we found nearly identical organellar DNA sequences in different SAG assemblies, we suspected a potential biological or technical contamination of these highly amplified sequences and decided to completely separate organellar sequences from the assemblies.

The presence of organellar scaffolds was searched using a combined approach. First a BLASTn analysis was done using scaffolds as queries against a database that contained all sequenced organelle genomes. Scaffolds similar to a known organelle genome (bit score >1000) were flagged. Then, a scaffold was considered to have an organelle origin if at least three predicted proteins from the scaffold showed similarities to proteins from the Curated Chloroplast Protein Clusters (CHL) or Curated Mitochondrial Protein Clusters (MTH) databases (http://www.ncbi.nlm.nih.gov/books/NBK3797/). Then, the two lists were merged. The scaffolds that were inferred to have come from organelles were retrieved from the SAG dataset for subsequent analysis and the corresponding proteins were removed from the nuclear protein dataset.

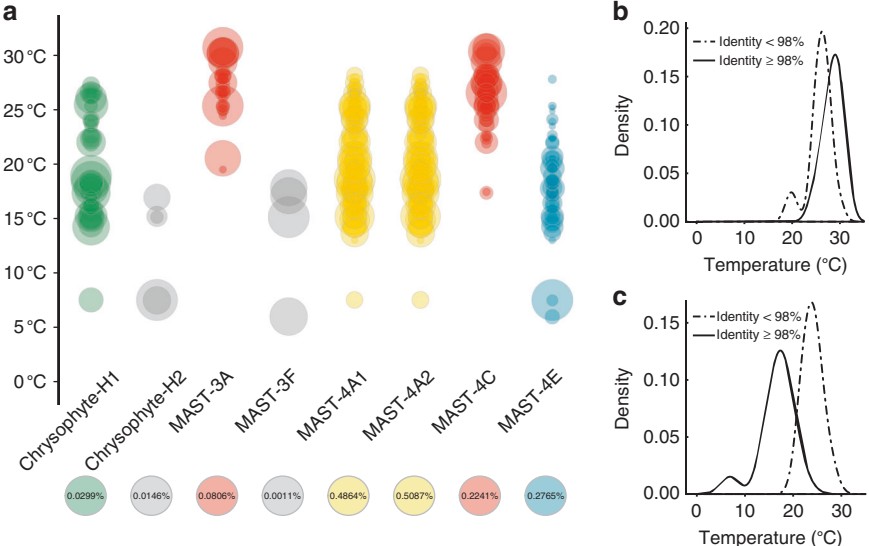

**Fig. 4** Water temperature and distribution of the heterotrophic protists. **a** x-axis represents the lineage composite genomes, and y-axis represents surface temperatures in degrees Celsius at each sampling location. Relative abundances are represented by circle size (one per station/depth where the genome was detected). The scale for each column is indicated below the name of the lineage. **b** MAST-3A abundance distribution relative to temperature. A difference in the distributions is observed with a p-value $<2\times10^{-2}$ (Wilcoxon test). **c** MAST-4E abundance distribution relative to temperature. Means are statistically different with a p-value $<3\times10^{-4}$ (Wilcoxon test). In **b** and **c**, line type indicates median sequence similarity to the reference genome assembly

**Cross-genera contamination removal**. To detect identical scaffolds in the distantly related SAGs from this study, all scaffolds were cut into 1000 bp-long fragments along a 500-pb overlapping sliding window. We used entire sequences of scaffolds shorter than 1000 bp. We aligned these fragments on each target assembly with BLAT and kept alignments with ≥95% identity >80% length. For each assembly, we considered and discarded contigs with at least one selected match with a distant phylum as contaminants. We distinguished three taxa: chrysophytes, MAST-3, and MAST-4. Subsequently, we assessed assembly completion using the BUSCO v2 pipeline[39] with the eukaryotic set of genes.

**Gene prediction**. Protein-coding genes were predicted by combining alignments of proteins from a custom database built from Uniref100 and MMETSP, alignments of transcripts from the *Tara* Oceans collection and ab initio gene models. The combination step was performed using the GAZE framework.

The custom protein database was based on Uniref100, with the addition of curated translated CDS from MMETSP transcripts and in-house sequenced transcriptomes. The final dataset contained more than 26 million proteins that were aligned using a two-step strategy. Protein sequences were first aligned using the fast BLAT program and significant matches were then re-aligned using the more accurate Genewise v2.2.0 software.

Transcripts from the *Tara* Oceans metatranscriptomic dataset were mapped using BLAST + 2.2.28. Significant alignments were then refined using est2genome, in particular to properly define exon–intron boundaries. To select organism-specific transcripts and avoid false positives, we only retained transcripts with ≥95% identity and with ≥80% of their length aligned onto the assembly.

Ab initio models were predicted using SNAP (v2013-02-16) trained on complete protein matches. Because of the insufficient number of complete proteins matching the MAST-3 F assembly, SNAP was trained on MAST-3 A assembly before running on that of MAST-3 F (Supplementary Table 6).

GAZE framework was used to integrate these three types of resources, using different weights to reflect their reliability. The most reliable resources—transcript alignments—were weighted 6.0, whereas protein alignments were weighted 4.5 and ab initio models 1.0. The weight acts as a multiplier for the score of each resource to build the final gene structure. Gene predictions with a GAZE score ≥0 were selected.

**Bacterial decontamination**. Bacterial scaffolds were detected using the alien index (AI)[40] calculated on each predicted gene. The alien index was defined as log(best eukarytic hit e-value + $10^{-200}$)−log(best non-eukaryotic hit e-value + $10^{-200}$). Thus, purely eukaryotic genes have a negative value whereas prokaryotic genes have a positive value. Scaffolds with predicted genes having an AI > 45 exclusively were considered as bacterial scaffolds and discarded from the final assembly.

**Metagenomic sequencing and mapping**. We sequenced 122 samples (accession numbers and contextual data in Supplementary Data 1–3) from 76 stations from the 0.8 to 5 μm size fractions (the size fraction where the studied MAST lineages

are most abundant), and obtained a total of $23.1\times10^{9}$ Illumina 101-bp paired-end reads. Reads from the 0.8 to 5 μm fraction size samples were mapped, in a three-step pipeline. In order to avoid the computation-intensive mapping of all reads, we first selected reads with at least one 25-mer in common with the target assembly. We then mapped the selected reads using bowtie2 2.1.0 aligner[41] with default parameters. Finally, we filtered alignments that correspond to low complexity regions using the DUST algorithm: alignments with <95% mean identity or <30% of high complexity bases were discarded.

**Discarding contaminants through metagenomic signatures**. The presence of unrelated sequences in the assembly was analyzed using a combination of approaches to obtain a list of scaffolds with atypical or suspect content. First, eukaryotic and prokaryotic signatures were determined for each scaffold. For this, a BLASTx analysis was conducted using the predicted gene as query against the nr-prot database (e-value threshold $<1\times10^{-0.5}$) followed by taxonomic assignment of each hit. A scaffold was determined to have a eukaryotic signature if it presented either at least one prediction assigned to one eukaryotic organism or none of the gene predictions had any similarities in the database. The scaffolds without these signatures were removed from the dataset. Second, we developed a new method to identify a population of scaffolds that co-vary in representation in the metagenomic data (see details below). This method identified outlier and inlier genes. The outlier dataset included genes with atypical behavior relative to the whole population of genes. Scaffolds that belonged to the outlier dataset were discarded. Supplementary Fig. 1b depicts an example of two different outlier scaffold groups (red), compared with the inlier scaffolds (blue). The three approaches were combined, which facilitated generation of a cleaned scaffold dataset and a corresponding cleaned gene dataset.

**Gene functional analysis: comparison of Pfam domain content between stramenopile genomes**. CDD search 3.11 was used for functional annotation of SAG genomes. Annotation was conducted on the cleaned gene dataset (see above) including outlier genes contained within single-gene scaffolds. We retrieved the Pfam motifs from CDD search output. Multiple occurrences of the same Pfam motif in one protein were counted as one. To perform a comparative analysis of the Pfam signature in the stramenopile taxa, we retrieved the protein dataset of representative available stramenopile genomes. To homogeneize these datasets from different projects, functional annotation of these gene datasets was performed. Proteins with similarities to CHL and MTH clusters were retrieved from the prior analysis. Because genome completeness was not similar between SAG lineages, random sorting of 1400 Pfam domains was independently performed 10 times for each genome. This threshold was selected because 1414 was the lowest number of Pfams, found per genome. A matrix with Pfam motif occurrence for all stramenopiles (10 random samplings per organism) was obtained. To visualize differences between Pfam content in stramenopile communities, we used non-metric multi-dimensional scaling (NMDS) based on Bray–Curtis dissimilarity distance. Bray–Curtis was used instead of Pearson correlation factor, because Bray–Curtis is unaffected by the addition or removal of Pfam motifs that are not

present in two gene repertoires. Moreover, it is unaffected by the addition of a new genome in the analysis. If Euclidean distance measures were used, the presence of double zeros in Pfam matrix abundance data may result in two genomes without any Pfam motifs in common being found to be more similar than other genome pairs with shared motifs. Bray–Curtis calculation and NMDS were created using the vegan package (v1.17-11) in R. Ellipses (95% confidence limit) were drawn in vegan using the ordiellipse function, with each group defined by common life history mode.

**Phylogenomic analysis.** The maximum likelihood phylogenetic tree of sequenced stramenopiles was reconstructed from conserved eukaryotic proteins detected using the BUSCO v2 pipeline. A total of 160 protein sequences present in at least four SAG assemblies were aligned using MUSCLE v3.8.31. Alignments were manually inspected to remove non-orthologous proteins (false positive detection with BUSCO). Subsequently, they were trimmed with Gblocks v0.91b using more relaxed parameters than default ($-b4=5$ $-b3=4$). Remaining trimmed sequences were concatenated. Because the selected 160 proteins were not present in all genomes, missing sequences were replaced by gaps ('-', character). Thus, the effective number of sequences used to infer phylogeny was often much lower than 160 (Chrysophyte H2: 51; MAST-4D: 72; MAST-3F: 73; MAST-3A: 88; MAST-4C: 90; MAST-4A1: 113; Chrysophyte H1: 113; MAST-4E: 115; MAST-4A2: 115). Phylogeny was inferred using RAxML v8.2.9 under the GAMMA model of heterogeneity in evolutionary rates among sites and using the JTT substitution model. Branch support was evaluated using 100 bootstrap pseudoreplicates.

**CAZyme analysis.** Using BLASTp[42], each encoded protein model was compared to the proteins listed in the CAZy database[24] (http://www.cazy.org/). Proteins with >50% identity over the entire domain length of an entry in CAZy were directly assigned to the same family, whereas proteins with 15–50% identity to a protein in CAZy were all manually inspected, aligned, and searched for conserved features, such as catalytic residues. Functional prediction was performed by BLASTp comparison of the candidate CAZymes against a library constructed with only the biochemically characterized CAZymes reported in the CAZy database under the 'characterized' tab of each family[43].

**HGT detection.** The presence of putative HGT events was determined using two methods. First, in the AI method[40], the 'inlier' gene dataset was used to query nr-prot (April 2014 version), and the BLASTx search output was used to calculate the AI. Additionally, a second step was also added to the AI method because the AI calculation is made using the first best hit from eukaryotes and prokaryotes: If a gene is wrongly assigned as prokaryotic, it would be erroneously considered an HGT event (false positive). Alternatively, if a closely related organism with a common HGT event is present in the database used for the BLAST search, a gene could be excluded from the putative HGT list (false negative). Consequently, the first 1000 hits were retrieved, taxonomically assigned, and classified in eukaryotic and prokaryotic classes. We considered genes with an AI > 45, predicted internally on a scaffold with more than five predicted genes as putative HGTs.

To validate these putative HGTs, we constructed a phylogenetic tree of the predicted protein and its 200 best BLASTp matches (Supplementary Data 4), but only allowing a maximum of three matches from the same genus to extend the sampled diversity. If less than 10 eukaryotic sequences were present in the 200 best BLAST matches, we included the 10 closest eukaryotic matches of all BLAST matches (8000 max). Sequences were aligned using MUSCLE 3.8.31 and non-conserved positions were discarded using GBlocks 0.91b with relaxed parameters ($-b3=10$ $-b4=5$ $-b5=h$). Phylogeny was inferred using RAxML 8.2.9 with JTT model and gamma model of rate heterogeneity ($-m$ PROTGAMMAJTTX parameter). We considered the tree to support the horizontal transfer hypothesis if the investigated gene did not cluster with other eukaryotic sequences (bootstrap value >50). In the other case, the putative HGT was eliminated and considered as a False Positive of the alien index method.

**Annotation of bacterial enzymatic activities in HGT.** A functional classification of HGTs was obtained using Intepro and Pfam motifs, and functional categories were determined using COG. The HGT protein sequences were used for protein-versus-protein alignments, using the BL2 option (BLAST allowing gaps) and a BLOSUM62 score matrix against UniProtKB. Those that had >30% identity over at least 80% of the length of the smaller of two compared sequences were kept. The best hit for each HGT was then selected. For each best hit, Interpro and Pfam classification identifiers were retrieved using the UniProtKB interface. Each HGT protein was then manually assigned to one functional category (cellular process and signaling, information storage and processing, metabolism, or poorly characterized) using their best hit functional annotation and signatures.

**Biogeography inlier/outlier detection.** The measurement of an organism's relative abundance from short-read metagenomic information is very difficult, because some genes may be highly homologous to orthologous genes from others organisms and attract cross-mapping metagenomic reads. Here, we present a statistical approach to discriminate genes with atypical mapping behavior. This analysis relies on the assumption that the values of the metagenomic RPKM (number of mapped reads per gene (intron plus exon) per kb per million of mapped reads) per gene follow a normal distribution. The presence of genes with mapping values distant from the majority of genes could have numerous causes, such as (i) presence of a scaffold coming from another organism, (ii) cross mapping, or (iii) genes with a high copy number. Outlier presence was determined using the Grubb's test. The test was conducted for a station if at least 20% of the organism's genes were detected. A gene was considered detected if at least one read mapped with 95% identity on 100% of the read length. The outlier lists for each station were merged to provide the outlier gene list. This detection allowed clear discernment of genes usable for relative abundance measurement (the inlier dataset) from unusable genes with noisy or random signal (the outlier dataset). Organism abundance measurements across stations is highly dependent on this filter (Supplementary Fig. 9a, b, f, and g), necessary for this type of analysis. However, the abundance measured in one station resulted from the combination of inlier and outlier genes (as in station 89 and 85 at surface, Supplementary Fig. 9c). The high number of stations sampled during the Tara Oceans expedition allowed us to show that outlier genes were detected in a large number of stations, which is expected for non-specific signals (Supplementary Fig. 9d, e).

**Biogeographic distributions.** Genes detected as outliers were removed from the biogeographic analysis. The relative abundance of an organism was measured as the sum of the number of mapped reads per gene divided by the total number of reads sequenced per station. Because only genes and not intergenic regions were used, a correction factor was applied to the relative abundance values: corrected relative abundance = raw relative abundance × assembly size/(size of the mapped genome × genome completion). The abundance in a geographical area was calculated as the mean of the relative abundance of all stations in the corresponding geographical area (Atlantic Ocean, Mediterranean Sea, Indian Ocean, Southern Ocean, and Pacific Ocean). For the world maps (e.g., Fig. 3), and to compare the SAG lineage abundance and reveal common patterns of occurrence, the data were normalized by dividing the relative abundance by the maximal relative abundance per organism. The world maps were generated using the R packages maps_2.1-6, mapproj 1.1-8.3, gplots_2.8.0, and mapplots_1.4.

**Correlations to environmental parameters.** We tested whether the SAG lineage presence and/or abundance in Tara Oceans samples were correlated with local physico-chemical conditions. We used physico-chemical parameter values obtained from each sampling site during the expedition, which are available in the PAN-GAEA database[27]. For each parameter, we performed a Kruskal–Wallis one-way test and a post-hoc Tukey's test. We statistically delineated SAG lineage classes. Only stations for which we detected at least 20% of genes from each composite assembly lineage were considered. MAST-3F was not present at a sufficient number of stations and was therefore excluded from statistical analyses.

**Code availability.** Computer code used to perform comparative genomics, calculate relative abundances and represent biogeographies is available from the corresponding authors upon request.

**Data availability.** Sequencing data are archived at ENA under the accession number PRJEB6603 for the SAGs (see Supplementary Table 5 for details) and PRJEB4352 for the metagenomics data (see Supplementary Data 3). All other relevant data supporting the findings of the study are available in this article and its Supplementary Information files, or from the corresponding authors upon request.

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

## Acknowledgements

We thank the commitment of the following people and sponsors who made this singular expedition possible: CNRS (in particular Groupement de Recherche GDR3280), European Molecular Biology Laboratory (EMBL), Genoscope/CEA, the French Governement 'Investissement d'Avenir' programs Oceanomics (ANR-11-BTBR-0008), FRANCE GENOMIQUE (ANR-10-INBS-09), MEMO LIFE (ANR-10-LABX-54), PSL* Research University (ANR-11-IDEX-0001-02), Fund for Scientific Research—Flanders, VIB, Stazione Zoologica Anton Dohrn, UNIMIB, ANR (projects 'PHYTBACK/ANR-2010-1709-01', POSEIDON/ANR-09-BLAN-0348, PROMETHEUS/ANR-09-PCS-GENM-217, TARA-GIRUS/ANR-09-PCS-GENM-218), EU FP7 (MicroB3/No. 287589, IHMS/HEALTH-F4-2010-261376), ERC Advanced Grant Award to CB (Diatomite: 294823), US NSF grant DEB-1031049 to M.E.S. and R.S., FWO, BIO5, Biosphere 2, agnès b., the Veolia Environment Foundation, Region Bretagne, World Courier, Illumina, Cap L'Orient, the EDF Foundation EDF Diversiterre, FRB, the Prince Albert II de Monaco Foundation, Etienne Bourgois, the *Tara* schooner and its captain and crew. *Tara* Oceans would not exist without continuous support from 23 institutes (http://oceans.taraexpeditions.org). We also acknowledge C. Scarpelli for support in high-performance computing. This article is contribution number 63 of *Tara* Oceans.

## Author contributions

R.M., O.J., M.Si., C.d.V., and P.W. designed the study. P.W. wrote the paper with substantial input from S.M., Y.S., Q.C., V.d.B., E.K., C.B., D.I., R.S., R.M., B.H., O.J., M.S., S. Su., C.d.V., P.H. and M.B.S. C.D., M.P., S.K.L., S.Se., and S.P. collected and managed *Tara* Oceans samples. J.P. and K.L. coordinated the genomic sequencing. N.P., R.S., and M.S. conducted SAG generation and identification. S.M., Y.S., Q.C., E.P., M.W., J.L., V.L., J.F. M., R.L., V.d.B., M.Sa., R.M., J.M.A., B.H., and O.J. analyzed the genomic data. D.I. analyzed oceanographic data. *Tara* Oceans Coordinators provided a creative environment and constructive criticism throughout the study. All authors discussed the results and commented on the manuscript.

## Additional information

**Competing interests:** The authors declare no competing financial interests.

Yoann Seeleuthner[1,2,3], Samuel Mondy [1,2,3], Vincent Lombard[4,5,6], Quentin Carradec [1,2,3], Eric Pelletier[1,2,3], Marc Wessner[1,2,3], Jade Leconte[1,2,3], Jean-François Mangot[7], Julie Poulain[1], Karine Labadie[1], Ramiro Logares [7], Shinichi Sunagawa [8,9], Véronique de Berardinis[1,2,3], Marcel Salanoubat[1,2,3], Céline Dimier[10,11,12], Stefanie Kandels-Lewis[8,13], Marc Picheral[14], Sarah Searson[15], Tara Oceans Coordinators, Stephane Pesant[16,17], Nicole Poulton[18], Ramunas Stepanauskas [18], Peer Bork[8], Chris Bowler[12], Pascal Hingamp[19], Matthew B. Sullivan[20], Daniele Iudicone [21], Ramon Massana[7], Jean-Marc Aury [1], Bernard Henrissat[4,5,6,22], Eric Karsenti[12,15,16], Olivier Jaillon [1,2,3], Mike Sieracki[23], Colomban de Vargas[10,11] & Patrick Wincker[1,2,3]

[1]CEA - Institut de biologie François Jacob, GENOSCOPE, 2 rue Gaston Crémieux, 91057 Evry, France. [2]CNRS, UMR 8030, CP5706 Evry, France. [3]Université d'Evry, UMR 8030, CP5706 Evry, France. [4]Centre National de la Recherche Scientifique, UMR 7257, F-13288 Marseille, France. [5]Aix-Marseille Université, UMR 7257, F-13288 Marseille, France. [6]INRA, USC 1408 AFMB, F-13288 Marseille, France. [7]Department of Marine Biology and Oceanography, Institut de Ciències del Mar (CSIC), E-08003 Barcelona, Catalonia, Spain. [8]Structural and Computational Biology, European Molecular Biology Laboratory, Meyerhofstraße 1, 69117 Heidelberg, Germany. [9]Institute of Microbiology, Department of Biology, ETH Zurich, Vladimir-Prelog-Weg 4, 8093 Zürich, Switzerland. [10]CNRS, UMR 7144, Station Biologique de Roscoff, Place Georges Teissier, 29680 Roscoff, France. [11]Sorbonne Universités, UPMC Univ Paris 06, UMR 7144, Station Biologique de Roscoff, Place Georges Teissier, 29680 Roscoff, France. [12]Ecole Normale Supérieure, PSL Research University, Institut de Biologie de l'Ecole Normale Supérieure (IBENS), CNRS UMR 8197, INSERM U1024, 46 rue d'Ulm, F-75005 Paris, France. [13]Directors' Research European Molecular Biology Laboratory, Meyerhofstraße 1, 69117 Heidelberg, Germany. [14]Sorbonne Universités, UPMC Université Paris 06, CNRS, Laboratoire d'océanographie de Villefranche (LOV), Observatoire Océanologique, 06230 Villefranche sur Mer, France. [15]Department of Oceanography, University of Hawaii, 96815 Honolulu, Hawaii, USA. [16]PANGAEA, Data Publisher for Earth and Environmental Science, University of Bremen, 28359 Bremen, Germany. [17]MARUM, Center for Marine Environmental Sciences, University of Bremen, 28359 Bremen, Germany. [18]Bigelow Laboratory for Ocean Sciences, East Boothbay, ME 04544, USA. [19]Aix Marseille Univ, Université de Toulon, CNRS, IRD, MIO UM 110, 13288 Marseille, France. [20]Departments of Microbiology and Civil, Environmental and Geodetic Engineering, Ohio State University, Columbus, OH 43210, USA. [21]Stazione Zoologica Anton Dohrn, Villa Comunale, 80121 Naples, Italy. [22]Department of Biological Sciences, King Abdulaziz University, Jeddah, 21589, Saudi Arabia. [23]National Science Foundation, Arlington, VA 22230, USA. Yoann Seeleuthner and Samuel Mondy contributed equally to this work

## Tara Oceans Coordinators

Silvia G. Acinas[7], Emmanuel Boss[24], Michael Follows[25], Gabriel Gorsky[16], Nigel Grimsley[26,27], Lee Karp-Boss[24], Uros Krzic[28], Fabrice Not[11], Hiroyuki Ogata[29], Jeroen Raes[30,31,32], Emmanuel G. Reynaud[33], Christian Sardet[16,34], Sabrina Speich[35,36], Lars Stemmann[16], Didier Velayoudon[37] & Jean Weissenbach[1,2,3]

[24]School of Marine Sciences, University of Maine, Orono, Maine, 04469, USA. [25]Department of Earth, Atmospheric and Planetary Sciences, Massachusetts Institute of Technology, Cambridge, MA 02139, USA. [26]CNRS UMR 7232, BIOM, Avenue du Fontaulé, 66650 Banyuls-sur-Mer, France. [27]Sorbonne Universités, Paris 06, OOB UPMC, Avenue du Fontaulé, 66650 Banyuls-sur-Mer, France. [28]Cell Biology and Biophysics, European Molecular Biology Laboratory, Meyerhofstrasse 1, 69117 Heidelberg, Germany. [29]Institute for Chemical Research, Kyoto University, Gokasho, Uji, Kyoto 611-001, Japan. [30]Department of Microbiology and Immunology, Rega Institute, KU Leuven, Herestraat 49, 3000 Leuven, Belgium. [31]Center for the Biology of Disease, VIB, Herestraat 49, 3000 Leuven, Belgium. [32]Department of Applied Biological Sciences, Vrije Universiteit Brussel, Pleinlaan 2, 1050 Brussels, Belgium. [33]Earth Institute, University College Dublin, Dublin 4, Ireland. [34]CNRS, UMR 7009 Biodev, Observatoire Océanologique, F-06230 Villefranche-sur-mer, France. [35]Department of Geosciences, Laboratoire de Météorologie Dynamique (LMD), Ecole Normale Supérieure, 24 rue Lhomond, 75231 Paris Cedex 05, France. [36]Laboratoire de Physique des Océans, UBO-IUEM, Place Copernic, 29820 Plouzané, France. [37]DVIP Consulting, 92310 Sèvres, France

