## [Peer Review File · Nature Communications]

Editorial Note: This manuscript has been previously reviewed at another journal that is not operating a transparent peer review scheme. This document only contains reviewer comments and rebuttal letters for versions considered at Nature Communications. Mentions of prior referee reports have been redacted.

Reviewers' comments:

Reviewer #5 (Remarks to the Author):

This manuscript presents the analysis of several single amplified genomes (SAGs) from marine stramenopiles belonging to the clades MAST3, MATS4 and chrysophytes. Despite the genomes are rather partial (ca. 50% on average), the authors are able to identify a number of genes allowing them to corroborate functions usually attributed to these clades from more classical analyses. Overall, although the methodological part of the work is solid, [redacted] the driven conclusions are not particularly novel, being sometimes stretched to make the data say what they cannot. In my view, given the absence of truly innovative results, this manuscript would better fit a more specific genome or microbial ecology journal. Some specific criticisms follow.

- In their title, the authors claim that their study "reveals specific niche adaptations and underestimated diversity". For the diversity part, it has been already clear for the past 15 years that we are still far of having a complete picture of protist diversity, such that including this phrase in the title is trivial and does not add useful information. The claim about "niche adaptation" is more problematic. For ecologists, a niche is something else than an extremely broad lifestyle category such as heterotroph or photoheterotroph plus preferred optimal temperatures. The authors not only incur in language abuse using the term "niche" but do not provide any evidence showing niche adaptation. In order to do so, they should make population genomics/genetics studies, which are fully absent from this work.

- It is extremely surprising that the authors do not include in their comparison the genome of *Incisomonas marina*, the only culturable MAST3 stramenopile for which the genome is available (Derelle 2016, which they avoid to cite). They must include *Incisomonas* at the very least in their phylogenetic trees to show the relative position of their SAGs, and they should possibly compare the absent/present functions in comparison with this genome.

- *Solenicola setigera* (MAST3) is not in culture and it is logical that they do not include it in a genome comparative analysis. Yet, it would be useful to include *Solenicola* in a 18S rRNA gene tree together with environmental sequences for a reference [redacted].

- One interesting observation is the presence of proteorhodopsin in one MAST4 subclade. However, while proteorhodopsin allows the synthesis of extra ATP from light, it does not directly intervene in the carbon cycle, since these organisms are not photosynthetic and consequently do not fix carbon. The impact in the C cycle would be only via the tuning of their already assumed heterotrophic ability.

- Horizontal gene transfer. The authors have revised the number of HGT cases inferred to lower numbers after making phylogenetic trees. The trees should be provided as supplementary

material. The authors use as criterion that a gene does not cluster with other eukaryotic genes in their trees, but these are reconstructed with FastTree and no minimal bootstrap support is mentioned. Tree topology and interpretation might be affected by various factors, including poor phylogenetic signal, poor taxon sampling, used models of sequence evolution or poor node support, not to mention contamination. Unambiguous proof for HGT needs to be provided.

- Recruitment of SAGs in TARA metagenomes. The authors use this information to deduce geographical distributions. I agree that the TARA dataset is the most comprehensive available so far and a very interesting one. Yet, it is relatively low in resolution [redacted] and it has the inconvenient that different samples have been collected at different time points. Hence, deducing geographical distribution and, furthermore, the "niche" from TARA recruitment is highly problematic. For instance, the authors find a strong correlation of SAG distribution with temperature. Well, if TARA would sample again the same points e.g. 6 months apart (winter-summer), the geographical distribution concluded for the same SAGs could be completely different. Now the heat-loving strains would be found at different geographical locations from those deduced for the first dataset. Therefore, while this recruitment information is useful and potentially indicative, it is largely overinterpreted. Pinpointing the geographical distribution and ecological preferences for those SAGs will require more comparative analyses or the inclusion of other datasets (including existing punctual/time series studies).

Point-by-point response to the reviewers - October 2017

Each comment of the reviewer (C#) is reported, immediately followed by the corresponding reply (Reply #).

Referee #5

This manuscript presents the analysis of several single amplified genomes (SAGs) from marine stramenopiles belonging to the clades MAST3, MATS4 and chrysophytes. Despite the genomes are rather partial (ca. 50% on average), the authors are able to identify a number of genes allowing them to corroborate functions usually attributed to these clades from more classical analyses. Overall, although the methodological part of the work is solid, [redacted] the driven conclusions are not particularly novel, being sometimes stretched to make the data say what they cannot. In my view, given the absence of truly innovative results, this manuscript would better fit a more specific genome or microbial ecology journal. Some specific criticisms follow.

C1 - *In their title, the authors claim that their study “reveals specific niche adaptations and underestimated diversity”. For the diversity part, it has been already clear for the past 15 years that we are still far of having a complete picture of protist diversity, such that including this phrase in the title is trivial and does not add useful information. The claim about “niche adaptation” is more problematic. For ecologists, a niche is something else than an extremely broad lifestyle category such as heterotroph or photoheterotroph plus preferred optimal temperatures. The authors not only incur in language abuse using the term “niche” but do not provide any evidence showing niche adaptation. In order to do so, they should make population genomics/genetics studies, which are fully absent from this work.*

Reply 1. We understand that the word “niche” here is inappropriate and we propose another title: “Single-cell genomics of multiple uncultured stramenopiles reveals underestimated functional diversity across oceans”. This new title emphasizes the fact that the underestimated diversity revealed here concerns the proteome composition rather than the species diversity, which is known to be broad.

C2. - *It is extremely surprising that the authors do not include in their comparison the genome of *Incisomonas marina*, the only culturable MAST3 stramenopile for which the genome is available (Derelle 2016, which they avoid to cite). They must include *Incisomonas* at the very least in their phylogenetic trees to show the relative position of their SAGs, and they should possibly compare the absent/present functions in comparison with this genome.*

Reply 2. At this time (September 2017), the genome of *Incisomonas marina* is not publicly available, only the raw reads have been deposited into the Sequence Read Archive. The genome assembly and the whole dataset of predicted proteins could not be found neither in the public databases nor on the website of the laboratory. Only 306 protein sequences were analyzed and published in the supplementary data of Derelle *et al.* (2016). There was therefore no intention to avoid citing the Derelle *et al.* article, but we simply didn't use their partial data in the previous versions of this paper. On the referee's request, we used 54 of the 306 published protein sequences to include *Incisomonas marina* in the phylogenetic tree of Figure 1b of the new version of the manuscript. However, it is impossible to compare absent/present functions without a larger set of predicted proteins.

C3. - *Solenicola setigera (MAST3) is not in culture and it is logical that they do not include it in a genome comparative analysis. Yet, it would be useful to include Solenicola in a 18S rRNA gene tree together with environmental sequences for a reference [redacted].*

Reply 3. [redacted]

Otherwise, to follow the referee's suggestion, we have added an 18S rDNA gene tree to Supplementary Figures (Supplementary Figure 3 of the revised manuscript). The conclusions remain the same, as we find the same tree topology as previously published 18S rDNA trees for these organisms (Gomez, Moreira et al. 2011; Derelle, Lopez-Garcia et al. 2016).

C4. - *One interesting observation is the presence of proteorhodopsin in one MAST4 subclade. However, while proteorhodopsin allows the synthesis of extra ATP from light, it does not directly intervene in the carbon cycle, since these organisms are not photosynthetic and consequently do not fix carbon. The impact in the C cycle would be only via the tuning of their already assumed heterotrophic ability.*

Reply 4. We completely agree with the referee on this point, and we did not claim that proteorhodopsins directly intervene in the carbon cycle.

C5. - *Horizontal gene transfer. The authors have revised the number of HGT cases inferred to lower numbers after making phylogenetic trees. The trees should be provided as supplementary material. The authors use as criterion that a gene does not cluster with other eukaryotic genes in their trees, but these are reconstructed with FastTree and no minimal bootstrap support is mentioned. Tree topology and interpretation might be affected by various factors, including poor phylogenetic signal, poor taxon sampling, used models of sequence evolution or poor node support, not to mention contamination. Unambiguous proof for HGT needs to be provided.*

Reply 5. We followed the referee's recommendations and used RAxML to reconstruct more reliable trees and added it to the manuscript as Supplementary Information (Supplementary Data 3 of the new version of the manuscript). We removed 5 additional cases of candidate HGTs that were clustering with eukaryotic sequences in the RAxML trees (also with low bootstrap values). We further classified 17 additional candidate HGTs as 'ambiguous cases' because their phylogenetic trees were well supported using FastTree, but the bootstrap values were very low (< 50) using RAxML. The genes classified as 'ambiguous cases' are still potential HGTs, but as the phylogenetic tree does not strongly support the transfer, we removed these genes from the functional analysis. In consequence, we updated the functional analysis of the remaining candidate HGTs (supplementary tables 3 and 4 of the new version of the manuscript) but the conclusions remain identical.

We agree that the detection of horizontal gene transfers is challenging and we have performed several steps to avoid pitfalls. First, to exclude obvious contamination, we limited our analysis to genes with homology to prokaryotic sequences but integrated in scaffolds assigned to eukaryotes. However, because we used Multiple Displacement Amplification, this type of scaffold could result from a chimeric assembly of the SAG sequences and contaminant DNA. This is why we verified that the abundance pattern of HGT candidates in metagenomic data was identical to non-HGT pattern (see an example fig. response A of this letter). The co-occurrence of HGT candidates and non-HGT genes strongly suggests that these HGT candidates are part of the SAG genome, and do not result from a chimeric assembly. Moreover, some HGT candidate homologs are found in multiple SAG assemblies and the evolutionary history of these genes – after the HGT event – corresponds to the phylogenetic tree of species (fig. response B of this letter). This result excludes a contamination from an organism having the same geographical distribution as the SAG.

In our opinion, the variations in our tree topologies are not a significant problem because most of the trees (115/243) are built from prokaryotic sequences only, even when including all eukaryotic sequences found in the 8,000 best BLAST matches.

To conclude, we have multiple evidence showing that these HGT candidates do not come from contaminant DNA sequences and are integrated into the genomes. In most cases, HGT candidates have similarity with prokaryotic proteins only, supporting a clear prokaryotic origin.

Nevertheless, we toned down the claims concerning HGTs in the manuscript and now use the terms 'potential HGTs' and 'candidate HGTs' (lines 229, 427, 434, 437, 438, 447, 839, 840 of the new version of the manuscript).

Figure response A. Abundance patterns of 2 subsets of MAST-4 A 2's genes in *Tara* Oceans metagenomic data. (a) Random sample of 74 eukaryotic MAST-4 A 2's genes. (b) All 74 MAST-4A 2 HGT candidates. In (a) and (b), the curve represents the mean RPKM of each sample and the colored zone covers 95% of the RPKMs. Pearson's correlation coefficient between mean RPKMs: 0.997.

a

b

Figure response B. Phylogenetic trees of orthologous proteins acquired by Horizontal Gene Transfers (HGT). a. Phylogenetic trees of the three HGT orthologous proteins found in the four MAST-4 genomes. b. Phylogenetic trees of the 12 HGT orthologous proteins found in three MAST-4 genomes. Orthologous proteins were retrieved using BLAST best reciprocal hits ($e\text{-value} < 1.10^{-10}$). Sequences were aligned using muscle 3.8.245 and phylogeny was inferred using RAxML 8.2.9 when the number of sequences was greater than 3 (a) or FastTree otherwise (b). Trees were rooted using the midpoint method.

C6. - *Recruitment of SAGs in TARA metagenomes. The authors use this information to deduce geographical distributions. I agree that the TARA dataset is the most comprehensive available so far and a very interesting one. Yet, it is relatively low in resolution [redacted] and it has the inconvenient that different samples have been collected at different time points. Hence, deducing geographical distribution and, furthermore, the "niche" from TARA recruitment is highly problematic. For instance, the authors find a strong correlation of SAG distribution with temperature. Well, if TARA would sample again the same points e.g. 6 months apart (winter-summer), the geographical distribution concluded for the same SAGs could be completely different. Now the heat-loving strains would be found at different geographical locations from those deduced for the first dataset. Therefore, while this recruitment information is useful and potentially indicative, it is largely overinterpreted. Pinpointing the geographical distribution and ecological preferences for those SAGs will require more comparative analyses or the inclusion of other datasets (including existing punctual/time series studies).*

Reply 6. Based on the list of metagenomics samples maintained on the EBI Metagenomics website¹, open ocean metagenomics samples are still rare, and global time-series are even rarer. More importantly, most metagenomics studies present in databases target the prokaryotic size-fraction, thus providing only limited possibilities to explore the novel eukaryotes of this study. We added "from the 0.8-5 μm size-fraction of *Tara* Oceans dataset" (line 237 in the manuscript) to emphasize the use of a eukaryotic size-fraction for the biogeography study.

We agree that using time-series at the global scale would be ideal to exhaustively understand the biogeography of these organisms, but such datasets do not exist for protists at the global scale. The use of scattered data would only give local information about the dynamics of these organisms: abundance variations over time would be specific to each location and global patterns would likely change. On the other hand, we consider that as a first step our work provides a global picture of their distribution at a certain time and indicate the physico-chemical parameters that most probably explain this distribution. We intentionally did not discuss the details of each geographical distribution but compared global patterns (global/restricted distribution; abundant in surface/depth) to avoid overinterpretations.

References

- Derelle, R., P. Lopez-Garcia, et al. (2016). "A Phylogenomic Framework to Study the Diversity and Evolution of Stramenopiles (=Heterokonts)." *Mol Biol Evol* **33**(11): 2890-2898.
- Gomez, F., D. Moreira, et al. (2011). "Solenicola setigera is the first characterized member of the abundant and cosmopolitan uncultured marine stramenopile group MAST-3." *Environ Microbiol* **13**(1): 193-202.

¹https://www.ebi.ac.uk/metagenomics/projects/doSearch?searchTerm=&studyVisibility=ALL_PUBLISHED_PROJECTS&biomeLineage=&includingChildren=false&biome=MARINE&search=Search&startPosition=0

REVIEWERS' COMMENTS:

Reviewer #5 (Remarks to the Author):

The manuscript has been significantly improved. The title and the presentation of the work are better focused. However, I still do not clearly see what is the major impact that this work brings for plankton ecology or protist evolution. It appears a well-done descriptive analysis of a few planktonic stramenopiles based on single cell genomes, but some of the conclusions are generalities (e.g. uncovering functional diversity) or are not as solidly shown as the authors claim. For instance, the claim "we show that each sequenced genome or genotype has a specific oceanic distribution, principally correlated with water temperature and depth" is not solidly substantiated for several reasons, as previously discussed: only one data set for comparison, limited to a particular cell-size fraction, different time points compared, absence of replicates. From this perspective, it would be important that the authors recognize the limitations of the biogeographic distribution inferred from this single data set and tone down their assertions. At most, their data are suggestive.

The authors might have done a much better job at integrating the descriptive information produced in a well-framed ecological or evolutionary problem. Instead, the introduction and discussion are thin and vague. A clear example is offered by the last two sentences of the introduction. The first of them reads "It has been postulated that all of these lineages originated from a presumably autotrophic stramenopile ancestor, although lack of genome information has hindered understanding of the evolution of heterotrophy versus autotrophy within the stramenopiles". The first part of the sentence seems to contradict the second, which alludes to the opposite evolutionary pathway. It is not clear what the authors intend with this sentence. Is it to propose solving an evolutionary conundrum with the data obtained from SAGs? However, nothing is mentioned about this aspect in the rest of the manuscript. The following sentence of the introduction does not help either: "Assessment of the genes involved in the degradation of organic matter may thus be relevant for elucidating their roles in marine ecosystems and biogeochemical cycles". How does this connect to the previous sentence? The link is not obvious. The manuscript would greatly benefit from an improved contextual framework.

Response to the reviewer

Reviewer #5 (Remarks to the Author):

The manuscript has been significantly improved. The title and the presentation of the work are better focused. However, I still do not clearly see what is the major impact that this work brings for plankton ecology or protist evolution. It appears a well-done descriptive analysis of a few planktonic stramenopiles based on single cell genomes, but some of the conclusions are generalities (e.g. uncovering functional diversity) or are not as solidly shown as the authors claim. For instance, the claim “we show that each sequenced genome or genotype has a specific oceanic distribution, principally correlated with water temperature and depth” is not solidly substantiated for several reasons, as previously discussed: only one data set for comparison, limited to a particular cell-size fraction, different time points compared, absence of replicates. From this perspective, it would be important that the authors recognize the limitations of the biogeographic distribution inferred from this single data set and tone down their assertions. At most, their data are suggestive.

We thank the referee for his positive evaluation on the improvement of the manuscript. The limitations of the biogeographic analysis are addressed in a specific section of the Discussion, written based on the reviewer remarks. We would emphasize that the Tara Oceans data set has not a limitation due to the use of a particular size fraction, this is in fact a choice made after looking at all size fractions available in this data set and focusing on the one that contains most signal for all the studied organisms.

The authors might have done a much better job at integrating the descriptive information produced in a well-framed ecological or evolutionary problem. Instead, the introduction and discussion are thin and vague. A clear example is offered by the last two sentences of the introduction. The first of them reads “It has been postulated that all of these lineages originated from a presumably autotrophic stramenopile ancestor, although lack of genome information has hindered understanding of the evolution of heterotrophy versus autotrophy within the stramenopiles”. The first part of the sentence seems to contradict the second, which alludes to the opposite evolutionary pathway. It is not clear what the authors intend with this sentence. Is it to propose solving an evolutionary conundrum with the data obtained from SAGs? However, nothing is mentioned about this aspect in the rest of the manuscript. The following sentence of the introduction does not help either: “Assessment of the genes involved in the degradation of organic matter may thus be relevant for elucidating their roles in marine ecosystems and biogeochemical cycles”. How does this connect to the previous sentence? The link is not obvious. The manuscript would greatly benefit from an improved contextual framework.

This manuscript is not oriented towards resolving the evolutionary trajectory of the stramenopiles. In particular, the question of their autotrophic or heterotrophic origin could only be tackled by looking at basal organisms. We focused either on ecologically important organisms, as defined on the basis of this and previous studies. Their capacities to degrade organic matter are taken as an indicator of their diverse potential roles.